# Domain Feature Mapping with YOLOv7 for Automated Edge-Based Pallet Racking Inspections

**DOI:** 10.3390/s22186927

**Published:** 2022-09-13

**Authors:** Muhammad Hussain, Hussain Al-Aqrabi, Muhammad Munawar, Richard Hill, Tariq Alsboui

**Affiliations:** 1Department of Computer Science, School of Computing and Engineering, University of Huddersfield, Huddersfield HD1 3DH, UK; 2Department of Computer Science, COMSATS University of Islamabad, Islamabad 45550, Pakistan

**Keywords:** defect detection, deployment, rack damage, smart manufacturing, warehouse automation

## Abstract

Pallet racking is an essential element within warehouses, distribution centers, and manufacturing facilities. To guarantee its safe operation as well as stock protection and personnel safety, pallet racking requires continuous inspections and timely maintenance in the case of damage being discovered. Conventionally, a rack inspection is a manual quality inspection process completed by certified inspectors. The manual process results in operational down-time as well as inspection and certification costs and undiscovered damage due to human error. Inspired by the trend toward smart industrial operations, we present a computer vision-based autonomous rack inspection framework centered around YOLOv7 architecture. Additionally, we propose a domain variance modeling mechanism for addressing the issue of data scarcity through the generation of representative data samples. Our proposed framework achieved a mean average precision of 91.1%.

## 1. Introduction

Smart manufacturing, also known as Industry 4.0, refers to manufacturing functions that are integrated and collaborative, and process the ability to act on data in a timely manner. Focusing on the operation level, the computerization of certain tasks leads to data generation via communication protocols. The data are procured, processed, and utilized to improve the decision-making power. The acquisition of data and the ability to facilitate interconnectedness between various distributed operations within a manufacturing facility are succinctly seen as smart manufacturing.

Pallet racking is one of the most ubiquitous infrastructures found within logistic/distribution centers, warehouses, and many other types of manufacturing facilities. Its purpose is to hold stock in a safe manner ready to be shipped when required. Depending on the nature and scale of operations, pallet racking can be densely populated, containing thousands of pallets of considerable monetary value.

The structural safety of racking is of paramount importance in order to avoid damage accumulation resulting in collapsed racking, which costs the business financially and also puts human lives at risk. Collapsed racking can be a result of two factors. The first is incorrect installation and the second is the careless operation of forklifts near the racking whilst loading/off-loading. This research presents a framework for the automated detection and localization of racking defects via YOLOv7 architecture coupled with AprilTag-based localization.

### 1.1. Literature Review

The current literature concerning the inspection of pallet racking suggests a shortage of active research for this area; there are only a couple of publications concerning pallet racking. Hence, we broadened our literature review to focus on structural defect detection.

Farahnakian et al. [1] proposed an image segmentation framework for the detection of damage to pallet racking. As part of their research, the authors mentioned the unavailability of any open-source racking data, leading them to manually collect a custom dataset. In their proposed methodology, we observed that the authors subscribed to the implementation of a Mask RCNN network based on ResNet-101 architecture as an approach for feature extraction. They reported achieving a mean average precision (MAP) of 93.45% with respect to an Intersection over Union (IoU) of 50%. However, the computational implications as a result of the Mask RCNN were not presented. Due to the internal architectural composition of the Mask RCNN, the computational demand would be very high; hence, the model—regardless of high performance—would not be suitable for deployment directly onto an edge computing device with a limited computational capacity.

Dong et al. [2] presented an evaluation of computer vision (CV) techniques for structural-based health monitoring (SHM). The authors categorized SHM into two factions: local and global. The local class referred to the identification of defects such as delamination, cracks, and loose bolt detection. The latter class consisted of a structural behavior analysis, vibration-based serviceability, modal identification, and damage detection. The authors presented their findings by highlighting the fact that CV-SHM necessitates high-quality representative data to provide profitable results. In many applications, the requirement of a large data acquisition for model training is not feasible due to the lack of an infrastructure and other factors such as the resultant down-time. The paucity of data acquisition, especially when it comes to image data and more so from within a manufacturing facility, can be designated as a major obstacle, resulting in a slow growth of automated CV applications within the manufacturing industry.

Similarly, Zhu et al. [3] examined image processing techniques manipulated by developers and researchers for CV-SHM applications. Their review presented a list of the ‘inherent distinctive’ benefits of CV-SHM such as non-contact, a long distance, multiple object detection, and electromagnetic inference. Reviewing the constraints of the present CV-SHM technology, the authors cited that the majority of CV-SHM applications are constrained within the walls of academic laboratories. As a result of this approach, models developed for addressing quality inspection issues have a high risk of failing when put into production due to a lack of representative training/testing and risks of false generalization from inadequate models.

We have observed an increase in the implementation of CV-based applications across various sectors. This is a result of recent and continuous improvements in CV-based architectures along with the wide adoption of transfer learning.

Around a decade ago in 2012, AlexNet [4] was introduced by Hinton et al., presenting a graphics processing unit (GPU) for accelerated calculations. The research also introduced the ReLu activation function for reducing the model convergence time. Since then, the CV community has observed a flurry of new architecture introductions, including the popular GoogleNet [5], VGGNet [6], RCNN [7], Fast RCNN [8], and Faster RCNN [9]. The driving factors behind these developments were an iterative improvement in the detection accuracy, the utilization of lightweight architectures, and the optimization of the inference speed.

The architectures mentioned above are widely applied in industries such as medicine [10,11], renewable energy [12], and autonomous vehicles [13,14,15,16,17]. However, the majority of the development and use cases are limited to R&D in many businesses, including manufacturing [18]. One of the reasons for this is due to the computational load of the architectures that require specific GPU hardware for carrying out production-based inferencing [19]. Although cloud-based inference mitigates the issue of commissioning on-site GPU hardware, due to the cyber risks attached to IoT devices [20,21] as well as GDPR [22] compliance requirements, businesses are more vigilant toward cloud-based data transfer and processing. This has garnered an incremental interest in edge device model deployment, where internet connectivity is not always a necessity.

Based on the above premise, CV researchers are actively developing new lightweight architectures suitable for edge device deployment utilizing limited computational resources. The two most popular architectures in this regard are Single Shot Detector (SSD) MobileNet [23] and You Only Look Once (YOLO), focusing on edge device speed and accuracy.

Adibhatla et al. [24] implemented YOLOv2 architecture for the inspection and classification of defects found within printed circuit boards (PCBs). The architecture was trained with a large dataset containing 11,000 images that had been expertly annotated, outputting a detection accuracy of 98.79%. As an additional step, the authors could have implemented certain augmentation techniques for scaling the data in a representative manner and capturing the variance that may be found within other manufacturing facilities due to variations in production line configurations, for example.

In our previous work [25], we proposed the implementation of MobileNet-SSDV2 for the detection of vertical damage to pallet racking based on a custom dataset that was manually collected and annotated. The proposed solution was successful in providing an edge device-compatible network (deployed on Raspberry PI) that achieved a MAP of 92.7% @ and an IOU of 50%. This work is an extension of our previous research, as we expand on the defect categories from a single class (vertical damage) to multiple classes along with representative data scaling and defect localization.

Summarizing the literature, we have learnt that there is a dearth of research that focuses on architectural quantization for addressing deployment issues. For example, the subscription to the segmentation domain of the CV as opposed to object detection in [1] means that the trained architecture (Mask RCNN), regardless of the fact that it may achieve a high accuracy, would have to be hosted on a cloud or specific GPU hardware when being put into production. Furthermore, there is a lack of representative data scaling and variance introduction—a fundamental mechanism for countering data scarcity—as is the case for pallet racking, where no open-source data are available.

### 1.2. Paper Contribution

Our first contribution to automated pallet racking came in the form of a classifier detecting five different classes: horizontal, vertical, support, vertical damage, and support damage. We addressed the issue of data scarcity through selected image processing techniques appropriate for modelling the variance induced within various distribution centers, warehouses, and manufacturing facilities as a result of internal and external factors. The scaled dataset was used to train the recent YOLOv7 architecture with a focus on producing a highly generalized model that was able to differentiate between various racking states.

Secondly, we benchmarked the trained architecture performance not only on architectural and computational performance but also on post-deployment metrics such as frame-per-second (FPS), demonstrating the realistic inference speed of the trained architecture. Additionally, we proposed a complete deployment framework and deployed the developed architecture onto a jetson device. We resolved the issue of determining the location of the damaged racking leg through the use of AprilTags and a positive-inference window (PiW) mechanism.

## 2. Methodology

### 2.1. Data Procurement

To the best of our understanding, there is no open-source pallet racking dataset consisting of representative samples that can be used for developing automated defect detection classifiers for addressing damage detection in pallet racking. This potentially justifies the reason for the lack of active research in this field as data are the fundamental components for initiating any development work in AI-related applications. Extending our research from [25], we presented the first multiclass pallet racking dataset collated from a pool of local manufacturing firms hosting various types of pallet racking.

Figure 1 presents our operational mechanism for the raw data acquisition, accumulation, and filtration. The process of data collection was carried out at three different warehouses. Smartphones were utilized for recording videos of pallet racking at each warehouse. The videos from each warehouse were collated into a single access point via Dropbox. A video splitter was then implemented for procuring static images of the racking with a split rate of 1 FPS. The resultant images formed the original raw dataset.

### 2.2. Data Pre-Processing

After deriving the original dataset based on the data procurement strategy, the next step involved the annotating of the data. This was a fundamental step; its efficiency would have an impact on the performance of the model. If the bounding boxes around the classes of interest were too loosely defined, this could force the models to generalize on a false assumption. Conversely, very stringent bounding boxes could result in missing a section of the relevant class, again leading to the risk of false generalization during training. In order to balance the two risks, we decided to annotate based on close proximity. At the same time, due to stock being placed on the racking, certain images contained occlusions with respect to the class of interest. To address this, images containing an occlusion of a quarter of the class of interest were fully annotated, including the occlusion part (Figure 2A); with occlusions greater than a quarter of the class of interest (Figure 2B), only the apparent region of the class was annotated, as shown in Figure 2.

### 2.3. Data Augmentation

It was noticed during the frame splitting that many frames did not contain any pallet racking due to the movement of the smartphone around the warehouse. These images were filtered out of the dataset. The next phase within the data transformation involved the scaling of the dataset thorough representative augmentations in order to provide enough training data for the selected architecture to generalize on. Figure 3 presents the proposed augmentation strategy.

### 2.4. Device-Induced Variance Modelling

The first type of variance modelling was based on capturing the variance that could be caused by the Jetson Nano used for capturing and inferencing the racking images. The Jetson Nano was located on the adjustable bracket of the forklift, as shown in Figure 4.

Although the device was initially installed (with a magnetic mount) onto the adjustable bracket in a certain orientation, the orientation could be adjusted due to external factors such as stock off-setting the device position whilst loading/off-loading. Hence, to cater for this variance and enable the model to accurately inference regardless of the orientation, shift and rotation augmentations were applied. For this, a shift component was introduced denoting the shift factor in the form of ($S_{x},S_{y}$):(1)$$M=\left[\begin{array}{ccc} 1 & 0 & S_{x}\\ 0 & 1 & S_{y} \end{array}\right].$$

The matrix *M* was translated into an array before applying an affine transform, where *inp* was the input image, *otp* was output image, and *M* equaled the transformation matrix:(2)$$otp\left(x,y\right)=inp(M_{11}x+M_{12}x+M_{13},M_{21}x+M_{22}y+M_{23}).$$

As result of pixel shifting, the variance caused by the off-setting of the hardware device was captured. However, this type of processing eliminated the damaged regions of the racking in a few cases (when rack damage was located at the edge of the image). To facilitate the preservation of damaged racking in the images, center-based rotations were introduced. Habitually, the rotation of an image with respect to an angle (*θ*) would be reached via the matrix:(3)$$Mb=\left[\begin{array}{cc} cos\theta & -sin\theta \\ sin\theta & cos\theta \end{array}\right].$$

However, the aim here was to reference the center of the image for the rotation:(4)$$Mb=\left[\frac{\alpha \beta \left(1-\alpha \right)\cdot centre.x-\beta \cdot centre.y}{-\beta \alpha \cdot centre.x+\left(1-\alpha \right)\cdot centre.y}\right]$$
where *centre* is the rotation center (input image), *θ* is the rotation angle (degrees), and scale is the isotropic scale factor:(5)α=scale·cosθ, β=scale·sinθ.

Additionally, the placement of the device on the adjustable brackets of the forklift enabled the strategic coverage of the particular racking on which the forklift was operating. However, this also resulted in a continuous dynamic state for the device to operate in; i.e., inferencing on the state of the racking. Depending on the speed of the forklift, the image data captured by the device could be blurry. To cater for this type of variance, Gaussian Blur was introduced. The pixel-wise blurring of certain images was aimed at introducing a representative variance; i.e., due to varying lighting conditions and the hardware specifications of camera device, the model was provided with a richer training dataset to assist with generalization. Figure 5A presents a shifted racking image whilst Figure 5B presents a blurred image.

### 2.5. Environmental Variance Modelling

This architecture was aimed at pallet racking detection across a wide range of warehouses, distribution centers, and manufacturing facilities. It was plausible to expect that different locations would present varying external factors that the model-trained architecture would need to adapt to. For example, warehouse M may have an increased lux intensity (increased lighting) whereas warehouse B, located in another country, may have significantly less lux intensity. It is also possible that warehouse M may have variations in its lux intensity during the day–night shifts.

To model the lux variations into the training dataset, a random brightness adjustment was implemented. This was achieved by varying the pixel intensity of a given image within a globally predefined intensity mechanism varying between −11 and +11%. Figure 6A presents a high intensity generated sample whilst Figure 6B presents a simulation of an low intensity environment sample.

The scaled dataset consisted of 2094 samples. Table 1 presents the dataset post-splitting into training, validation, and test sets for facilitating the training and evaluation process.

### 2.6. YOLOv7 Architecture

There are many architectures available for training custom defect detection classifiers. We selected the most appropriate based on the following criteria. First, the research focused on defect detection without requiring pixel-level accuracy; hence, object detection was selected as opposed to segmentation. Second, object detection has many promising architectures; however, the benchmark here was real-time inference speed along with an acceptable accuracy. Based on the above two filters, the architecture search was narrowed down to MobileNet and the YOLO family of models. MobileNet was implemented in [25]; as an extension, YOLO was selected for this research. YOLO contains various variants with continuous improvements. However, with the recent advent of YOLOv7 [26] in 2022, we have provided the first racking detection application based on this architecture. Another reason for the selection of YOLOv7 was based on the paper claiming that the architecture was the fastest and most accurate real-time detector to date.

### 2.7. YOLOv7 Architectural Reforms

YOLOv7 presents several architectural reforms aimed at improving the detection speed and accuracy. In general, all YOLO architectures consist of a backbone, head, and neck. The backbone is responsible for the grounding work, extracting essential features and feeding them through to the head via the neck component. YOLOv7 moves away from its predecessors when it comes to the backbone; that is, rather than utilizing the darknet, an extended efficient layer aggregation network (E-ELAN) is deployed as the computational block for the backbone. Although the E-ELAN paper has not been published yet, the concept is based on the use of expand, shuffle, and merge cardinality to continuously enhance the learning ability of the network without losing the original gradient path.

Furthermore, to address the issue of model scaling for a specific device deployment, researchers generally utilize a Network/Neural Architecture Search (NAS) tool [27]. NAS enables a parameter iteration search to unearth optimal scaling factors based on the resolution, width, depth, and stage (the number of feature pyramids). For the YOLOv7 architecture model, the scaling is further enhanced through a compound model scaling mechanism. This is achieved by the coherent scaling of the width and depth parameters for concatenation-based models.

YOLOv7 also caters for re-parameterization planning (RP). RP is based on the concept of averaging various models to produce a final model that is robust in performance. Module-level re-parameterization has been an active area of research, where certain segments of the model have exclusive re-parameterization strategies. YOLOv7 utilizes the gradient flow propagation paths for determining the segments (modules) within the overall model that require re-parameterization. Finally, the head component of the architecture is based on the multihead concept. That is, the lead head is responsible for the final classification whilst the auxiliary heads assist with the training process in the middle layers. Figure 7 presents the abstract training framework for the pallet racking application.

## 3. Results

### 3.1. Hyperparameters

In order to initiate the training process, a set of hyperparameters had to be defined. Although the training could have been carried out via Google Colaboratory, the GPU allocation for the free tier is limited and, hence, timeout issues could result incomplete training. Hence, a standalone system was utilized for the complete training of the architecture. The system specifications along with the defined hyperparameters for guiding the training process are presented in Table 2.

### 3.2. Model Evaluation

Various metrics were utilized to comprehend the model performance from different perspectives and granularities. Along with the generic metrics of evaluation precision, recall, and F1 scores, the Intersection over Union (IoU) was specifically used because our application was based within the object detection realm as opposed to image classification or segmentation. The IoU, also referred to as the Jaccard Index, facilitated similarity quantifications between the ground truth $M_{g}$ and the predicted $M_{p}$ bounding boxes, as shown in (6):(6)$$\mathrm{IoU}=\frac{area\left(M_{p}\cap M_{g}\right)}{area\left(M_{p}\cup M_{g}\right)}.$$

$M_{p}$ and $M_{g}$ were predefined as 0.5, interpreted as a 50% overlap between the ground truth $M_{g}$ and the model prediction $M_{p}$; these must be satisfied for the model prediction to be classified as correct. Furthermore, MAP (mean average precision) was preferred due to its comprehension of the sensitivity of the model. Firstly, the precision, recall, and F1 score were computed for a predefined confidence threshold of 50%. The MAP was calculated (7), where $AP_{i}$ was the average precision for *i-th* class and *C* equated to the number of classes:(7)$$\mathrm{MAP}=\frac{1}{C}\;\sum_{i=1}^{C}AP_{i}.$$

Table 3 presents the overall performance of the YOLOv7 architecture. The trained architecture was able to achieve an impressive MAP@0.5-IoU of 91.1% at 19 FPS. Commenting on the convergence capacity of the architecture, it could be seen that this accuracy was achieved in approximately 6 h of training time.

The training time coupled with the resultant performance was a manifestation of the effectiveness of our proposed data scaling mechanism. As presented in Table 1, the dataset post-scaling was still small in conventional terms. However, due to the targeted modelling of representative augmentations, there was a sufficient underlying feature representation for the model to highly generalize.

Figure 8 provides a granular class-specific performance indicator in the form of a precision recall curve. From the p-r curve we observed that the vertical racking provided the highest performance (95.3%) followed by the vertical damaged racking (92.6%).

The support racking and damaged support racking saw a decrease in performance of 88.1% and 88.6%, respectively. This was due to the fact that the support racking was not as visually apparent as the vertical and horizontal racking as it was located on the sides of the vertical racking providing support; hence, it was usually obscured when loaded with stock. Thus, the limited class access to these support racking images could have resulted in a decrease in the generalizability of the model for this particular class. However, the overall performance of the architecture was still 91.1%.

## 4. Discussion

In order to evaluate the overall success of our research it was essential to provide a comparison against similar research conducted in present times. Table 4 presents a comparison of this research with [1,25], as discussed earlier.

A prior work [25] focused on object detection whilst [1] opted for image segmentation. Although the objective in all three cases was the same, we preferred the object detection jurisdiction over segmentation for various reasons.

First, our research objective was to train a lightweight architecture that could be deployed onto an edge device to provide a real-time inference. This could not be achieved using the approach taken by [1] as a Mask RCNN architecture based on the ResNet-101 backbone contains 44.5 million learnable parameters, making it unsuitable for deployment onto a constraint computational device. Additionally, revisiting the common goal of racking defect detection meant that pixel-level segmentation was not required to achieve this; using the IoU concept via object detection, we were able to set a threshold from the overlap between the annotated and the predicted damage for a correct classification.

With respect to the data size, [1] only contained 75 images, but provided the highest accuracy. However, when observing the dataset (as presented in Figure 9), it could be seen that the data were not representative of the operational environment; rather, close-up images of the racking were utilized. Conversely, our dataset was also considered small in conventional terms, but was more representative as it was collected within the production environment and then further scaled based on representative augmentations.

Although our trained architecture had ostensibly the lowest performance (91.1%), the difference was minimal compared with [1,25] and less than 3%. However, when looking at the number classes, we observed that our architecture inferenced on five different classes. If a comparison was undertaken with [25] based only on vertical damage, then the difference in performance would have been 0.1%.

Finally, we also presented the FPS for our trained architecture, achieving a satisfactory performance of 19 FPS. The FPS coupled with the detection accuracy based on the five different classes testified to the success of our proposed framework.

## 5. Conclusions

In conclusion, our research was successful in the development of the first multiclass pallet racking detection framework. The performance manifestation through the achieved MAP coupled with the real-time performance of the trained architecture (19 FPS) demonstrated that the proposed framework could be utilized to deploy the system within warehouses, distribution centers, and manufacturing facilities for the monitoring of pallet racking.

Our solution presented a non-invasive approach to defect detection that was distinct from conventional racking monitoring products aimed at sensor-oriented solutions [28]. These require the installation of a hardware device—essentially, an accelerometer—on every single racking leg, significantly increasing the costs for clients. Conversely, our solution was strategically located on the adjustable brackets of the forklift, providing coverage of the racking near where the forklift was operating. Hence, the number of devices required for our application would be proportional to the number of forklifts within the warehouse of the client as opposed to the number of racking legs, significantly reducing the upfront and continuous maintenance costs.

Furthermore, the proposed framework could also be applied to various other domains where the objective may be to reduce the reliance on hardware-oriented data collection and inference such as PV fault detection and post-system deployment [29,30].

## Figures and Tables

**Figure 1 sensors-22-06927-f001:**
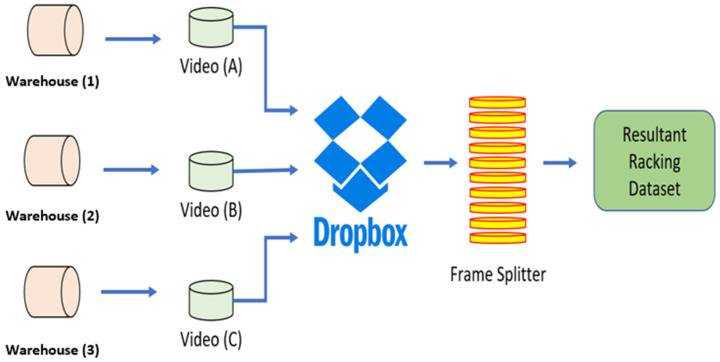
Data procurement strategy.

**Figure 2 sensors-22-06927-f002:**
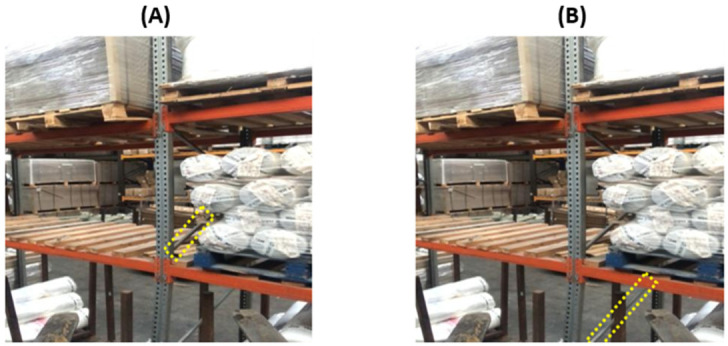
Data annotation strategy. (**A**) Higher occlusion (**B**) Small Occlusion

**Figure 3 sensors-22-06927-f003:**
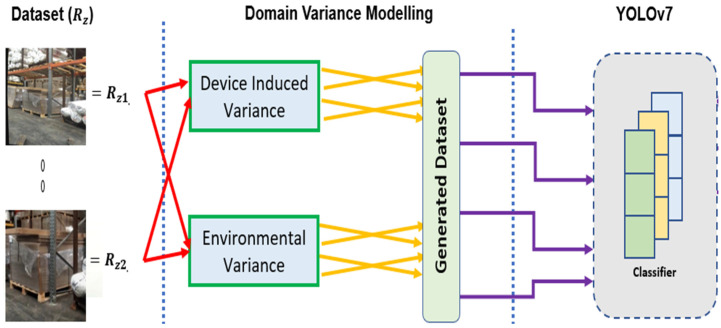
Variance modelling strategy.

**Figure 4 sensors-22-06927-f004:**
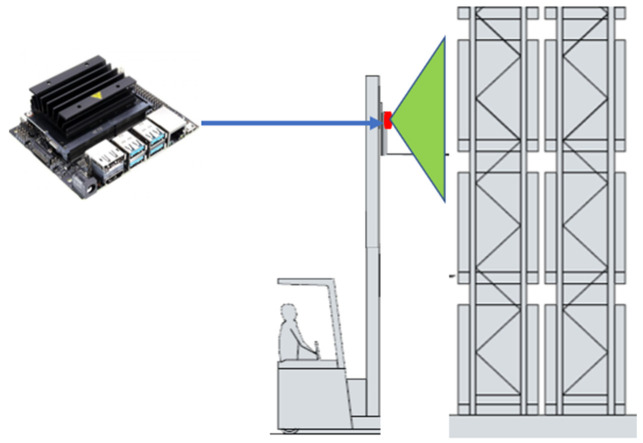
Strategy of device placement.

**Figure 5 sensors-22-06927-f005:**
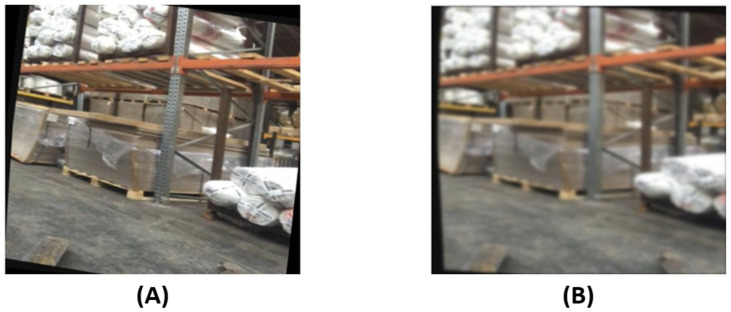
Data scaling. (**A**) Shifted Image (**B**) Implementing Gaussian Blur.

**Figure 6 sensors-22-06927-f006:**
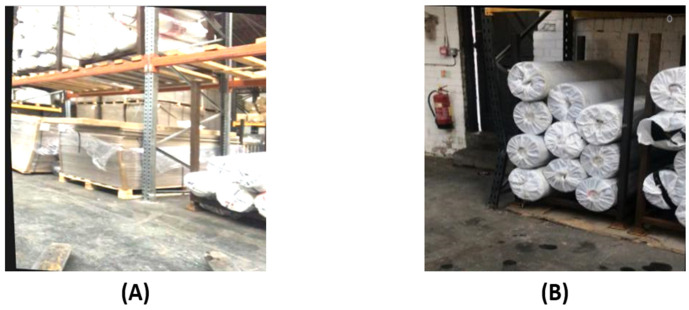
Domain specific augmentations. (**A**) High Intensity (**B**) Low Intensity.

**Figure 7 sensors-22-06927-f007:**
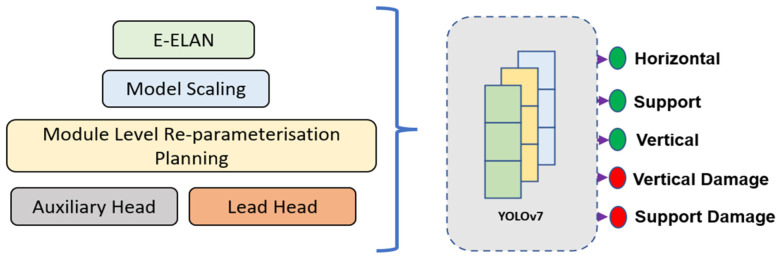
Proposed system architecture.

**Figure 8 sensors-22-06927-f008:**
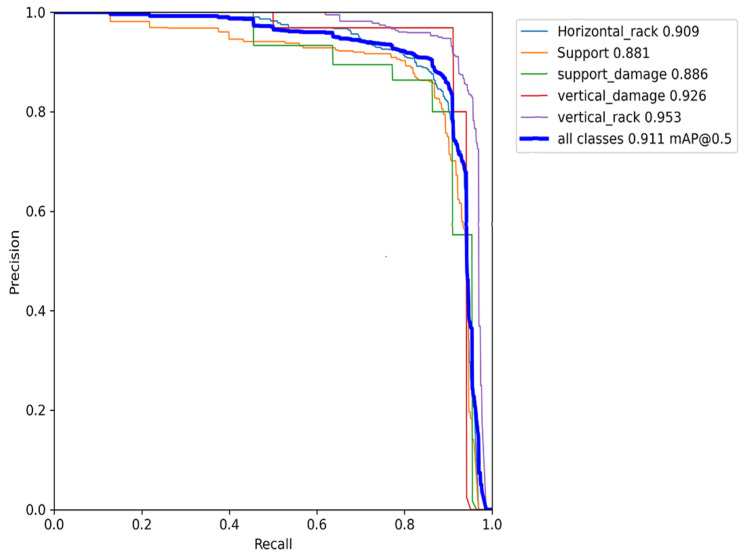
Precision recall curve for trained YOLOv7.

**Figure 9 sensors-22-06927-f009:**
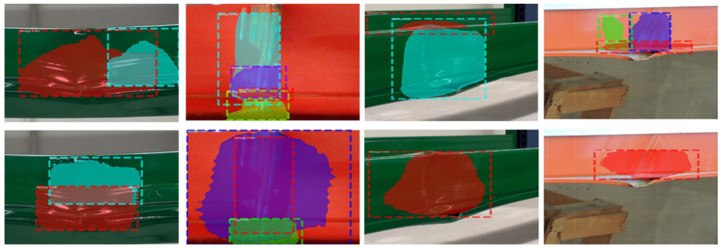
Data samples from [1].

**Table 1 sensors-22-06927-t001:** Transformed dataset.

Data	Samples
Training	1905
Validation	129
Test	60

**Table 2 sensors-22-06927-t002:** Hyperparameters.

Batch Size	20
Epochs	300
Optimizer	ADAM
Learning Rate	0.01
GPU Memory	5 GB
GPU	Quadro P2200

**Table 3 sensors-22-06927-t003:** Model evaluation.

MAP@50(IOU)	91.1%
FPS	19
Steps	300
Training Time	~6 h

**Table 4 sensors-22-06927-t004:** Recent work comparison.

	Our Research	Research by [1]	Research by [25]
Approach	Object Detection	Image Segmentation	Object Detection
Dataset Size	2094	75	19,717
Classes	5	1	2
Detector	YOLOv7	Two-Stage	Single Shot
MAP@0.5(IoU)	91.1%	93.45%	92.7%

## Data Availability

Not yet available.

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
