# Peer review of "Domain Feature Mapping with YOLOv7 for Automated Edge-Based Pallet Racking Inspections"

_sensors, 2022, doi:10.3390/s22186927_

Round 1

Reviewer 1 Report

Dear Authors,

This work presents very interesting autonomous rack inspection in order to guarantee stock protection and personnel safety. However, to raise the value of the article, please pay attention to these issues and make every effort to improve and supplement article.

1.      page 5 of 15 – "It was noticed during the frame splitting that many frames did not contain any pallet racking due to the movement of the smartphone around the warehouse."

Manuscript is missing the validation of video data or some kind of certificates for making videos of racks. There is no evidence that certified inspectors might be skillful to do their jobs with mobile phones.

2.      page 6 of 15 – equations are not numerated,

equation:

???(?,?)=???(?11?+?12?+?13,?21?+?22?+?23

seems unfinished

3.      page 7 of 15 – "The pixel-wise blurring of certain images would enable the model during training to generalize on this type of variance improving its chances of correct classification during post deployment inferencing".

Please, add reference for this claim, or provide proof.

4.      page 7 of 15 – "This application is aimed at pallet racking detection across a wide range of warehouses, distribution centers and manufacturing facilities."

What application? Please, explain what did you mean by word "application".

5.      page 8 of 15 – Section 2.4. named "Yolov7 Architectural Reforms" should be numerated as 2.7.

6.      Table 1 – sample test and validation test are pretty small compare to entire dataset. Please, explain why did you decide to make a test and validation with such a small validation and test datasets.

7.      page 9 of 15 "Furthermore, to address the issue of model scaling for specific device deployment, researchers generally utilize a Network Architecture Search (NAS) tool."

Which researchers? Please, give detail explanation about statement that researchers generally utilize a Network Architecture Search (NAS) tool or add required references.

8.      Overall remark: You didn’t discuss 6 h training time that was needed to achieve listed results, neither in the benchmark analysis, nor from the point of application of your work in real systems. So, please, make an effort to justify 6 h training time from the perspective of "prior works" (references [1] and [25]) that you used and from the perspective of applications in real systems

Kind regards

Author Response

Review Report 1

Point raised:

page 5 of 15 – "It was noticed during the frame splitting that many frames did not contain any pallet racking due to the movement of the smartphone around the warehouse."

Manuscript is missing the validation of video data or some kind of certificates for making videos of racks. There is no evidence that certified inspectors might be skillful to do their jobs with mobile phones.

Answer:

Firstly, thank you for the positive and constructive feedback.

The video and frame splitting process would not be part of the deployment infrastructure but rather part of the initial data collection strategy for the procurement of dataset. This was a fundamental step as mentioned in section 2.1, no open-source data was publicly available hence we had to collect our own data. Smartphones were used for data collection i.e., video recording of racking which were then processed as detailed in data processing/augmentations. This does not entail that a smartphone would be the end product for defect detection but rather the trained architecture could be deployed on existing cameras placed on forklifts or edge devices such as the jetson nano mentioned in the article.

section 2.1 states at the onset,

“To the best of our understanding there is no open-source pallet racking dataset consisting of representative samples that can be used for developing automated defect detection classifiers for addressing damage detection in pallet racking. This potentially justifies the reason for the lack of active research in this field as data is the fundamental component for initiating any development work in AI related applications. Extending our research from [25] we present the first multiclass pallet racking dataset collated from a pool of local manufacturing firms hosting various types of pallet racking.

Figure 1 presents our operational mechanism for raw data acquisition, accumulation and filtration. The process of data collection was carried out at 3 different warehouses. Smartphones were utilized for recording videos of pallet racking at each warehouse. The videos from each warehouse were collated into a single access point via dropbox. A video splitter was then implemented for procuring static images of the racking with a split rate of 1 FPS. The resultant images formed the original raw dataset.”

Point raised:

page 6 of 15 – equations are not numerated,

equation:

???(?,?)=???(?11?+?12?+?13,?21?+?22?+?23

seems unfinished

Answer:

Equations numbering has been addressed and the mentioned equation has been completed.

Point raised:

page 7 of 15 – "The pixel-wise blurring of certain images would enable the model during training to generalize on this type of variance improving its chances of correct classification during post deployment inferencing".

Please, add reference for this claim, or provide proof.

Answer:

Thank you for raising this point, we have rephrased this section to make our objective clearer. The aim of introducing augmentations is for variance injection and scaling. However, in our case we don’t arbitrarily introduce augmentations but rather provide the domain relevance. With the pixel-wise blurring, the aim was to capture variance that may occur as a result of variations i.e., Lux intensities from warehouse to warehouse, this would contribute towards the generalisation of the model as it would be learning from representative batches of data. Figure 5 is in itself proof that the selected augmentation was representative as it does not compromise the overall image structure but rather blurs the pixels to achieve the representations mentioned above. Rephrased section,

“The pixel-wise blurring of certain images was aimed at introducing representative variance i.e., due to varying lighting conditions, hardware specifications of camera device, providing the model with a richer training dataset to assist with generalization.”

Point raised:

page 7 of 15 – "This application is aimed at pallet racking detection across a wide range of warehouses, distribution centers and manufacturing facilities."

What application? Please, explain what did you mean by word "application".

Answer:

Thank you for pointing this out. It has been corrected to ‘architecture’ as the research is based on the development of a CNN architecture for racking damage detection.

Point raised:

‘ page 8 of 15 – Section 2.4. named "Yolov7 Architectural Reforms" should be numerated as 2.7.

Answer:

Thank you, this has been addressed.

Point raised:

Table 1 – sample test and validation test are pretty small compare to entire dataset. Please, explain why did you decide to make a test and validation with such a small validation and test datasets.

Answer:

Thank you. The progression and continues advancements with lightweight CNN architectures especially the Yolo family of object detectors focus more on data generalisation via representative samples as opposed to the sheer size. Furthermore, the collection of custom data due to lack of open-source availability also limited our size but the focus of the research was on addressing this through representative scaling. There are many cases where limited data samples are used, and the focus is on augmentation and regularisation for model generalisation such as;

Automatic detection of photovoltaic module defects in infrared images with isolated and...by Akram, M. Waqar; Li, Guiqiang; Jin, Yi ; More... Solar energy, 03/2020, Volume 198

Point raised:

 page 9 of 15 "Furthermore, to address the issue of model scaling for specific device deployment, researchers generally utilize a Network Architecture Search (NAS) tool."

Which researchers? Please, give detail explanation about statement that researchers generally utilize a Network Architecture Search (NAS) tool or add required references.

Answer:

A relevant reference has been added [27] for this.

Point raised:

 Overall remark: You didn’t discuss 6 h training time that was needed to achieve listed results, neither in the benchmark analysis, nor from the point of application of your work in real systems. So, please, make an effort to justify 6 h training time from the perspective of "prior works" (references [1] and [25]) that you used and from the perspective of applications in real systems

Answer:

The 6-hour training time would not have any implications on the model post deployment as the training is used within the development stage only. Therefore, the training time was not mentioned within the comparison as it has no implications, post deployment. Also training time does not impact the accuracy or inference after deployment as the optimal weights are frozen hence it becomes irrelevant as far as post deployment within the production site is concerned.

Reviewer 2 Report

The authors present a multi-class pallet racking detection framework based on YOLOv7.

The Literature review is not just literature review but contains also background. These should be in separate subsections.

Figure 1 needs improvement as Factory (A) should be replaces by Warehouse(1), (2) and (3).

It is not clear how the parameters of Table 2 are selected. Some explanation should be given. 

There are no results showing how the training and validation converge. These should be supplied. 

The dataset is small and overfitting might be occurring. 

Author Response

Review Report 2

Point raised:

The Literature review is not just literature review but contains also background. These should be in separate subsections.

Answer:

Firstly, thank you for the positive comments and constructive feedback. We decided to keep this section under literature review as the inclination of the review was towards image processing and hence most of the review was based on actual developments outcomes and CNN architectures.

Point raised:

Figure 1 needs improvement as Factory (A) should be replaces by Warehouse(1), (2) and (3).

Answer:

Thank you, this has been addressed.

Point raised:

It is not clear how the parameters of Table 2 are selected. Some explanation should be given.

Answer:

Hyperparameter’s are utilised in all CNN training and are globally defined for maintain integrity. For example, the number of epochs would mandate the training iterations and ADAM which is the de facto optimiser for weight updation would facilitate fundamental concepts such as backpropagation. As stated in section 3.1, Google Colab was not used, one of the reasons for this was timeout issues with the free subscription. Hence our own system was used for training with the key parameters for training presented in Table 2.

Point raised:

There are no results showing how the training and validation converge. These should be supplied. 

Answer:

As this research was focused on object detection as opposed to image classification, the MaP was the de facto metric for gauging the performance of the model as this would directly assess the model predictions against the ground truth labels. This is presented in Table 3. Furthermore, the PR curve provides a class breakdown performance of the architecture presented in Figure 8.

Point raised:

The dataset is small, and overfitting might be occurring. 

Answer:

Thank you. The progression and continues advancements with lightweight CNN architectures especially the Yolo family of object detectors focus more on data generalisation via representative samples as opposed to the sheer size. Furthermore, the collection of custom data due to lack of open source availability also limited our size but the focus of the research was on addressing this through representative scaling. There are many cases where limited data samples are used, and the focus is on augmentation and regularisation for model generalisation such as;

Automatic detection of photovoltaic module defects in infrared images with isolated and...by Akram, M. Waqar; Li, Guiqiang; Jin, Yi ; More... Solar energy, 03/2020, Volume 198